# Extracellular Vesicles: The Next Generation of Biomarkers and Treatment for Central Nervous System Diseases

**DOI:** 10.3390/ijms25137371

**Published:** 2024-07-05

**Authors:** Gabriele Zanirati, Paula Gabrielli dos Santos, Allan Marinho Alcará, Fernanda Bruzzo, Isadora Machado Ghilardi, Vinicius Wietholter, Fernando Antônio Costa Xavier, João Ismael Budelon Gonçalves, Daniel Marinowic, Ashok K. Shetty, Jaderson Costa da Costa

**Affiliations:** 1Brain Institute of Rio Grande do Sul (BraIns), Pontifical Catholic University of Rio Grande do Sul (PUCRS), Porto Alegre 90610-000, RS, Brazil; paula.gabrielli@acad.pucrs.br (P.G.d.S.); fernanda.bruzzo@edu.pucrs.br (F.B.); isadora.ghilardi@edu.pucrs.br (I.M.G.); vinicius.wietholter@edu.pucrs.br (V.W.); fxavier@pucrs.br (F.A.C.X.); joao.goncalves@edu.pucrs.br (J.I.B.G.); daniel.marinowic@pucrs.br (D.M.); jcc@pucrs.br (J.C.d.C.); 2Institute for Regenerative Medicine, Department of Cell Biology and Genetics, Texas A&M University School of Medicine, College Station, TX 77807, USA; ash.shetty@tamu.edu

**Keywords:** exosomes, ectosomes, extracellular vesicles, diagnostic, sequencing, biomarkers

## Abstract

It has been widely established that the characterization of extracellular vesicles (EVs), particularly small EVs (sEVs), shed by different cell types into biofluids, helps to identify biomarkers and therapeutic targets in neurological and neurodegenerative diseases. Recent studies are also exploring the efficacy of mesenchymal stem cell-derived extracellular vesicles naturally enriched with therapeutic microRNAs and proteins for treating various diseases. In addition, EVs released by various neural cells play a crucial function in the modulation of signal transmission in the brain in physiological conditions. However, in pathological conditions, such EVs can facilitate the spread of pathological proteins from one brain region to the other. On the other hand, the analysis of EVs in biofluids can identify sensitive biomarkers for diagnosis, prognosis, and disease progression. This review discusses the potential therapeutic use of stem cell-derived EVs in several central nervous system diseases. It lists their differences and similarities and confers various studies exploring EVs as biomarkers. Further advances in EV research in the coming years will likely lead to the routine use of EVs in therapeutic settings.

## 1. Background

Extracellular vesicles (EVs) constitute a heterogeneous range of membrane-bound vesicles, broadly classified as exosomes and microvesicles. Exosomes originate from endosomes, whereas microvesicles are formed by the direct budding of the plasma membrane [1]. The formation of exosomes depends directly on the biogenesis of multivesicular bodies (MVBs) [2]. These membrane structures are highly dynamic and directly involved in internalizing extracellular ligands or cellular components, promoting their recycling or degradation [2,3]. Endosome formation occurs directly from the invagination of the plasma membrane [1,4]. While most endosomes undergo lysosomal degradation, some will mature and become late endosomes [1,5], which generate intraluminal vesicles (ILVs) through the internal budding of the endosomal membrane itself. During this event, nucleic acids, lipids, and cytosolic proteins are incorporated into the vesicles formed [1]. The ILVs within late endosomes range in size from 30 to 140 nm and mature as exosomes [1,3,6,7]. This process begins with the reorganization of the endosome membrane so that it is highly enriched in tetraspanins [8]. Among tetraspanins, CD9 and CD63 are critical in exosome biogenesis [1]. Subsequently, the endosomal sorting complexes necessary for transport (ESCRTs) are recruited to the ILV formation site [9]. It is known that the generation of ILVs is facilitated by the interaction between syntenin and ALIX and the availability of heparan sulfate, syndecans, ALIX, and ESCRTs [10]. Although the ESCRT pathway is considered the primary mode of generation of exosomes, different studies have shown other ESCRT-independent mechanisms for exosome biogenesis [11,12]. Eventually, the direct fusion of the MVB with the plasma membrane promotes the release of exosomes into the extracellular space [1]. In facilitating exosome release, RAB11 and RAB35 play an auxiliary role in the fusion of the MVB with the plasma membrane [13,14]. However, another mode of exosome release may be via the budding of the plasma membrane independently of Rab GTPases [1]. Another way is through SNARE proteins [15,16], typically involved in the membrane fusion process between two organelles [17,18]. In conclusion, exosomes have varied biogenesis and release of different endosome subtypes through various mechanisms and may contain different internalized molecules depending on the cell type and physiological state.

The formation processes of microvesicles [19] (MVs) are less known than exosomes [1,20]. They are believed to involve vertical molecular traffic directly to the plasma membrane, associated with a redistribution of membrane lipids and the use of machinery on the surface to allow the compression and formation of the vesicle [20,21]. Therefore, a series of processes result in the biogenesis of MVs, with the redistribution of phospholipids, including repositioning phosphatidylserine in the plasma membrane and the contraction of the actin–myosin machinery [4]. For example, external factors may contribute to the release of MVs, such as the influx of calcium inducing the redistribution of membrane phospholipids, ultimately resulting in the increased release of MVs [22].

The International Society for Extracellular Vesicles (ISEV) has played a crucial role in advancing the field of EV research by providing guidelines and recommendations for EV classification [23]. ISEV guidelines categorize EVs into three main subtypes based on their biogenesis pathways: exosomes, MVs, and apoptotic bodies [23].

In addition to biogenesis, EVs can also be classified based on their size, which is typically determined using techniques such as nanoparticle tracking analysis (NTA), dynamic light scattering, or electron microscopy. While exosomes generally range in size from 30 to 150 nm in diameter, MVs are typically larger, ranging from 100 to 1000 nm, and apoptotic bodies can be even larger, exceeding 1000 nm [24]. It is important to note that EVs cannot be classified into exosomes or microvesicles based on their size or cargo composition. Therefore, the International Society for Extracellular Vesicles guidelines suggest the EV scientists refrain from using the phrase “exosomes” unless there is evidence they have been released from endosomes [23]. Instead, the use of the phrases small EVs (sEVs) and large EVs is recommended based on the size of EVs. Here, we use the general term “EVs” to refer to small EVs, unless otherwise specified.

Furthermore, ISEV guidelines recommend characterizing EVs based on their molecular composition, including protein, lipid, nucleic acid, and metabolite content. This involves profiling EV cargo using techniques such as Western blotting, mass spectrometry, RNA sequencing, and lipidomics to identify specific biomarkers and functional molecules associated with EVs [23]. Overall, the ISEV guidelines provide a comprehensive framework for classifying extracellular vesicles based on their biogenesis, size, and molecular composition. By adhering to these guidelines, researchers can ensure consistency and comparability across studies, facilitating the advancement of EV research and the translation of findings into clinical applications.

## 2. EVs as Key Mediators of Intercellular Communication

Cellular communication is fundamental for coordinating various biological processes, ranging from development and tissue homeostasis to immune responses and disease progression [25,26]. While direct cell–cell interactions through membrane-bound receptors and soluble mediators have been extensively studied, recent research has unveiled the crucial role of EVs in mediating intercellular communication [27].

EVs encapsulate a diverse array of biomolecules, reflecting the composition and physiological state of their parent cells. Proteomic, lipidomic, and nucleic acid analyses have revealed that EV cargo encompasses proteins involved in cell signaling, membrane trafficking, and cytoskeletal organization, as well as nucleic acids such as microRNAs, mRNAs, and DNA fragments [28,29,30,31,32]. Importantly, the selective packaging of cargo into EVs is governed by specific sorting mechanisms, including protein post-translational modifications, RNA-binding proteins, and lipid raft domains [33].

Upon release into the extracellular milieu, EVs can interact with neighboring or distant cells to modulate their biological functions [26]. The uptake of EVs by recipient cells occurs through a variety of mechanisms, including endocytosis, membrane fusion, phagocytosis, and receptor-mediated internalization [34]. Once internalized, EV cargo molecules can exert diverse effects on recipient cells, influencing gene expression, cell signaling pathways, and cellular behavior [26,35].

EVs in the CNS are primarily derived from neurons, astrocytes, microglia, oligodendrocytes, and endothelial cells of the neurovascular unit [36,37,38]. Neuronal-derived EVs contain synaptic proteins, neurotransmitters, microRNAs, and mitochondrial components, reflecting their role in synaptic communication and neuronal homeostasis [39,40,41]. Neurons and glial cells can internalize EVs released by neighboring or distant cells, leading to functional changes in recipient cells [42]. Importantly, EV uptake is influenced by cell type-specific receptors, membrane lipid composition, and extracellular microenvironment cues.

Despite the evolving knowledge, the study of EVs is hindered by several factors. Primarily, the fields in which the mentioned elements are studied face the following issues: Isolation and Purification: Due to the small size and heterogeneity, isolating and purifying EVs from different biological fluids remain problematic. Ultracentrifugation, several size-exclusion chromatography methods, immunoaffinity-based procedures, etc., have already been developed, each having its strength and limitations [41,43]; Characterization: Due to the need to define their size, concentration, morphology, surface markers, and cargo contents comprehensively, the identification of EVs remains rather complex. Previously used methods, include nanoparticle tracking analysis, electron microscopy, flow cytometry, mass spectrometry, etc.; however, the techniques lack standardization and validation [41,44]; Functional assays: Understanding their biological function and identification of cargo molecules requires developing functional assays. However, the major experimental limitations remain in establishing cause–effect relationships between EVs and cellular responses, defining the mechanisms underlying cargo delivery, and analyzing EV-mediated signaling [45]. Also, there are other challenges related to loading efficiency, structural stability, and defining exosome origins [46]. However, the first attempts to overcome these issues have already been achieved.

## 3. Potential and Applications of EVs-Derived Mesenchymal Stem Cells

Recent studies are exploring the therapeutic value of mesenchymal stem cell-derived extracellular vesicles (MSC-EVs) for treating infectious diseases [47]. These results suggest that MSC-EVs can eliminate pathogens, regulate immunity, and secrete antimicrobial factors, repairing tissue injuries, inhibiting the replication of pathogens, and activating macrophages [47]. Previously, the therapeutic efficiency of MSCs was attributed to their capability to migrate and engraft in damaged tissues [48]. However, studies showed that administered MSCs typically do not reach the target tissue in sufficient numbers. On the other hand, the secretome of MSCs, including EVs, growth factors, and cytokines, likely reaches the various organ systems [48]. Such observations have led to the theory that the efficacy of MSC treatments mainly comes from the paracrine effects mediated by the secreted factors of MSCs [47]. MSCs have gained much importance in the past decades due to their therapeutic and clinical roles [42,48]. However, recent studies have highlighted the importance and advantages of using MSC-EVs because of their ability to cross biological barriers, higher safety profile, and lower risks for tumorigenesis [48].

Several studies demonstrated the deep involvement of EVs in the physiology of the central nervous system (CNS), including processes such as synaptic activity, neuron–glia communication, and immune system response [49]. As is well known, since MSC-EVs can cross biological barriers, such as the blood–brain barrier (BBB), to modulate the immune response, repair damaged tissue, and reduce inflammation, MSC-EVs seem attractive for treating CNS diseases.

MSC-EVs can transport RNA, proteins, and other biologically relevant molecules from cell to cell; EVs can also transport viruses, viral proteins, and nucleic acids [50]. A study conducted by Wenshuo Zhou [51] showed that MSC-EVs can carry Zika virus, suggesting the possibility of disease spreading in CNS by EVs. In the context of Zika virus infection and, perhaps, other infections, studies in this area are fundamental to understanding the importance of EVs and developing strategies to treat CNS infections using EV-based drugs [51].

In addition to their excellent utility in infection treatments, EVs may be useful as disease biomarkers. EVs of various cells can be isolated from peripheral circulation using different methods [52]. In a study conducted by Pulliam [52], brain-derived EVs purified by a precipitation/immunoaffinity approach using antibodies against the neuronal cell adhesion molecule L1CAM demonstrated the composition of EVs in CNS disorders, including HIV-associated neurological disorders (HANDs) and Alzheimer’s disease (AD) [52]. Thus, the characterization of EVs is a valuable tool to identify biomarkers for CNS diseases, including infections. Therefore, exploring the possibilities of treating CNS infections with MSC-EVs is vital. The advantages of using EVs over the use of MSCs are apparent. Due to the properties of crossing physiological barriers, such as the BBB, reduced risks of immune response, not replicating after intravenous or intranasal administration, and being less likely to form clots, they seem ideal for treating brain disorders [53].

## 4. Central Nervous System as a Target and Niche for EVs

The importance of intercellular communication in the body’s physiology is well established. Recent research has revealed that EVs can modulate and coordinate signal transmission in the brain [54]. According to studies, practically every cell in the CNS may release EVs, allowing them to play various roles in the brain [55], such as mediating communication between neurons and glial cells [56,57]. They can also facilitate proximal and distant communication between cells in the CNS and the periphery [58].

The brain’s microvascular system is necessary for maintaining homeostasis. EVs generated by vascular endothelial cells (ECs) have been reported to increase the survival, proliferation, and migration of oligodendrocyte precursor cells [59]. An essential component of the brain is the neuron, an electrically excitable cell that communicates with other cells via synapses. Additionally, research has revealed that neurons can communicate with other cells by producing EVs. These neuron-derived EVs (nEVs) can carry miRNA into ECs, specifically miR-132, to regulate the integrity of the vasculature by targeting the eukaryotic elongation factor-2 kinase (eek2) [60]. Furthermore, growth factors such as the vascular endothelial growth factor (VEGF) and fibroblast growth factor (FGF) may be present in these nEVs, which can be released into the extracellular matrix (ECM) and contribute to protection [61]. nEVs can also be shed from glutamatergic synapses, which are likely to bind to nearby neurons and cause interneuronal communication [62,63]. An intriguing study discovered that nEVs can be released in situ, enhancing neuronal activity and modulating astrocyte function [64]. The secretion of nEVs can also be controlled via depolarization [65,66]. In addition, they can increase excitatory transmission at the presynaptic location by increasing the lipid fraction, such as using sphingolipid metabolism in neurons [67]. Therefore, nEVs can modulate neuron function, but neurons can modulate EV function [64], demonstrating that communication can go both ways [68]. Furthermore, these nEVs can also deliver miRNA into astrocytes, causing the glutamate transporter-1 (GLT1) protein to be regulated [59]. nEVs can also communicate with microglia by increasing the expression of the C3 protein, which causes neurite degeneration [60]. This suggests that the nEVS can regulate synaptic function by inducing microglia phagocytosis, causing weaker synapses to be removed while solid synapses are retained [60]. Furthermore, EVs can stimulate neurite development [61] as well as contribute to cell survival by lowering apoptosis-induced cell death [62].

Glia cells in the CNS are responsible for the inflammatory response and neurotropic support [63]. More specifically, research indicates that microglia—a subset of glial cells—are primarily responsible for communication with neurons [58]. Furthermore, microglia play an important role in the initial response to pathological processes such as injury and infection [64]. Aside from their role in brain physiology, reactive microglia can release an miR-146a-5p that inhibits two synaptic function proteins through EVs, which can propagate the degenerative process in a neuronal culture, resulting in a reduction in dendritic spine and synapse function [65].

The release of EVs into the cerebrospinal fluid (CSF) has been found to be influenced by age [66]. This research adds to an in vitro investigation that found astrocyte-derived EVs (ADEVs) could transport oligodendrocyte senescence-related factors, including arginase-1 and tyrosine phosphatase zeta [67]. Astrocytes form a thick arborization that supports CNS cells and helps in regulating chemical impulses. As a result, ADEVs can carry miRNAs, which regulate neuronal protein expression and alter dendritic arborization in hippocampus neurons [68]. Also, they can transmit Neuroglobin (NGB) to neurons, which reduces degeneration and hence contributes to protection [69]. They can also produce growth factors that help cells in the brain to differentiate [70]. ADEVs can impact the activity of neural stem progenitor cells (NSPCs), which produce many cell types in the brain. They can attach to NSPCs and induce differentiation into a specific phenotype, such as astrocytes. As a result, these EVs can control the crossover between neurogenic and gliogenic phases during development [71]. EVs containing inflammatory factors like tumor necrosis factor-alpha (TNF-a) are not limited to microglia [72]. Astrocytes can also release proinflammatory molecules like IL-1b in the brain [73]. ADEVs have a distinct signature when released by reactive astrocytes. ADEVs comprise a protein associated with target selectivity, such as integrins, as well as a cytokine associated with neuroinflammation [74]. Neurotoxic proteins may be present, resulting in neurodegeneration [75]. Reactive astrocytes can shed an increased quantity of EVs to aid tissue healing [76].

Oligodendrocytes are CNS cells that support, maintain, and mediate chemical signals that can change information. In addition, myelin is produced in the axons of mature oligodendrocytes, which is required for action potential transmission [77]. Studies have shown that EVs play a part in all of this communication by maintaining axonal integrity [78] and sending non-useful cellular material to microglia, which helps to retain myelin [79]. Furthermore, by lowering oxidative stress, oligodendrocyte-derived EVs can improve neuronal metabolism and protect neurons [55]. Increases in Ca2+ in the microenvironment can help to preserve myelin by stimulating the release of myelin-associated proteins such as myelin proteolipid protein (PLP), 2′,3′-Cyclic nucleotide 3′-phosphodiesterase (CNP), and myelin basic protein (MBP) [80]. The fusion of EVs generated from oligodendrocytes in the axons can be improved by increasing Ca^2+^ [80].

Overall, EVs can play an important role in maintaining brain homeostasis, but they can also be shed by injured cells, contributing to developing pathological conditions in the CNS [81]. As a result, the question arises as to whether such EVs are involved in CNS disorders.

## 5. EVs as Biomarkers for CNS Diseases

EVs, secreted by virtually all cells [82], can protect their encapsulated material from the extracellular environment [37] and cross the blood–brain barrier in either direction [83]. As a result, EVs released by CNS cells during a disease condition may reflect the altered physiology of the secretory cell [84]. These EVs, detected in biological fluids such as the CSF, blood, urine, and saliva, could be biomarkers [85] as shown in Figure 1.

Despite advances in finding CNS disease biomarkers based on blood and CSF samples and neuroimaging techniques, some limitations remain. In this context, EVs and their cargo, such as proteins and miRNAs, are being investigated as potential biomarkers in CNS diseases, particularly in peripheral samples. Such investigations can contribute to developing sensitive and non-invasive biomarkers for diagnosis, prognosis, and disease progression [86,87,88]. There are clinical trials using EVs as biomarkers for CNS diseases, such as Parkinson’s (NCT04603326; NCT05320250), Huntington’s (NCT06082713), TBI (NCT05279599), and stroke (NCT05370105). The studies’ characteristics and main findings are summarized in Table 1.

As it will be discussed in the following topics, EVs content can be used as biomarkers of several neurological conditions such as Alzheimer’s disease (AD), Parkinson’s disease (PD), Huntington’s disease (HD), multiple sclerosis (MS), epilepsy, traumatic brain injury (TBI), and amyotrophic lateral sclerosis (ALS). It is well known that EVs carry lipids, proteins, peptides, and nucleic acids such as mRNA, miRNA, and DNA [12]. Such cargo is what makes the EVs useful tools as biomarkers. Since each cell and tissue type express different molecular signatures in both health and disease, the EVs act as a mark of the cellular condition [79]. As EVs are much less invasive than conventional biopsies, they represent a major advantage for the diagnosis and prognosis of several neurological disorders.

## 6. Alzheimer’s Disease

Alzheimer’s disease (AD) is a progressive neurodegenerative disease and the most common cause of dementia. Amyloid-β and tau protein aggregates are the principal neuropathological hallmarks of AD [109]. However, there is no correlation between the quantities of these proteins measured in the CSF and blood. In patients with AD, blood levels of the brain-derived exosomal proteins amyloid-β42 (Aβ-42), total-tau (t-tau), tau phosphorylated at threonine 181 (p-T181-tau), and tau phosphorylated at serine 396 (p-S396-tau) are greater than in control patients. These findings are consistent with what is seen in the CSF, indicating that they could be used in clinical practice as non-invasive premortem AD biomarkers [155]. Still, in patients with AD and mild cognitive impairment, neurogranin levels are higher in the CSF and plasma but lower in the blood plasma neuronal-derived EVs [106]. However, combining the EV-associated synaptic proteins neurogranin, growth-associated protein 43 (GAP43), synaptosome-associated protein 25 (SNAP25), and synaptotagmin 1 could potentially predict AD 5 to 7 years before cognitive impairment [89]. Recently, EVs proteome suggested a connection between cathepsin B and AD pathogenesis [94].

Numerous miRNAs transported by EVs associated with AD have been described in the literature [91,146,156,157]. The fact that the Let-7 miRNA is enriched in nEVs when detected in plasma suggests that it could be a biomarker for AD diagnosis [158]. Yang et al. found that serum EVs enriched with miR-135a and miR-384 and depleted with miR-193b are characteristics of AD patients compared to controls. EV-associated miR-384 was the most effective in distinguishing between other dementia diseases among the three miRNAs [93]. Whether or not these isolated findings are promising, the profile of miRNAs carried by EVs in different biochemical fluids in AD patients allows for a systemic assessment of the process [159,160,161].

As a biomarker discovery tool, EVs can be used to track the prognostic and progression of AD. According to a pilot study conducted by Li et al., hsa-miR-1306-5p, hsa-miR-93-5p, hsa-miR-424-5p, and hsa-miR-3065-5p, as well as the expression of P-S396-tau in plasma-derived EVs, provide a protein and miRNA signature to differentiate AD patients from other dementia disorders and healthy patients. Therefore, it could afford a better prognostic capacity [162]. Aharon et al. investigated the role of EVs in the progression of AD. As the disease progressed, the neural markers MOG and CD171 in serum EVs increased proportionally. EVs’ inflammatory cytokines were elevated in AD patients. Furthermore, a link between the severity of the disease and the lowering of particular miRNAs was discovered. Thus, this conjunction of alterations could be employed as a diagnostic for AD severity [92].

## 7. Parkinson’s Disease

Parkinson’s disease (PD) is a neurological disease that causes motor dysfunction [96]. The etiology and spread of the disease are attributed to the aggregation of alpha-synuclein (α-syn), which in part may be mediated by EVs [163]. Although α-syn levels in EVs have been investigated as potential biomarkers for the premature diagnosis and progression of PD, many findings are conflicting. Patients with PD have a greater α-syn concentration in CNS-derived plasma EVs, which is associated with disease severity [164]. A larger concentration of α-syn in plasma EVs could distinguish PD from control patients and other illnesses that can produce α-syn aggregation [97], corroborating the previous conclusion. In individuals with early-stage PD, the levels of α-syn in plasma neuronal exosomes are increased and longitudinally greater than baseline levels [98]. In addition, the rise in α-syn in nEVs also occurs before the diagnosis of PD and continues to increase with disease progression [86]. These findings could help to promote early diagnostic biomarkers and prognostic for PD progression [98,99].

Patients with mild and severe PD, on the other hand, have much lower plasma levels of α-syn EVs than patients without PD [165]. Furthermore, in patients with PD, the level of α-syn in CNS-derived serum exosomes is smaller than in patients with essential tremor and healthy people. When patients with PD were subdivided, the decline was more significant in non-tremor-dominant (NTD) patients than in tremor-dominant (TD) patients. Although inconsistent with other findings, this could help to promote PD diagnostic biomarkers and distinguish NTD from TD [166]. These discrepancies could be due to variances in extraction processes, types, and markers from particular CNS cells from EVs. Therefore, α-syn deserves additional studies to develop as a useful clinical biomarker. In addition, an increase in α-synuclein (α-synOlig) and α-synOlig/total α-synuclein ratio in EVs was seen in alternative fluids such as saliva. Therefore, this fluid can be used as an alternative to promote α-syn-associated PD biomarkers [102]. Recently, a study showed that plasma EVs contain elevated synaptic proteins which can be candidates as biomarkers for the clinical progression of PD [109]. In plasma-derived EVs from PD patients, the expression of hsa-miR-30c-2-3p and miR-331-5p was increased, while the expression of hsa-miR-15b-5p, hsa-miR-138-5p, hsa-miR-338-3p, hsa-miR-106b-3p, and hsa-miR-431-5p and miR-505 was decreased [108]. While miR-24 and miR-195 had an increased expression in PD patients’ serum EVs, miR-19b had a lower expression [110]. Furthermore, a profile of six serum EV-derived miRNAs expressed differently in PD patients can be employed as early PD diagnostic and progression biomarkers [111]. miR-34a-5p, which is involved in neurobiology, was upregulated in plasma small EVs and associated with disease duration [112].

A proteomic analysis of EVs in the serum, urine, and plasma possibly derived from neurons revealed that several proteins are differentially present between PD patients and controls [100,167,168,169]. The DJ-1 protein is significantly elevated in EVs derived from plasma neurons as well as exosomes isolated from urine in PD patients compared to controls [104,105]. Furthermore, Ser(P)-1292 LRRK2 levels are high in idiopathic PD patients’ urine exosomes [106]. The significantly higher plasma EV pro-IL-1 and TNF- levels in PD patients compared to controls could indicate that inflammation is involved in the disease’s development and progression [113].

## 8. Huntington’s Disease

Huntington’s disease (HD) is a neurodegenerative hereditary condition that usually results in motor, cognitive, and psychiatric disorders [170]. It is caused by the repeat expansion of the CAG in the huntingtin gene (HTT). The mutant HTT (mHTT) protein includes an expanded polyglutamine (polyQ) repeat and can form aggregates, which are neurotoxic [170]. In vitro, EVs were found to integrate both extended repeat RNA and the polyQ protein. Despite this, no evidence of neurotoxicity has been found, most likely due to experimental limitations [171]. Platelets have a higher concentration of mHTT than other blood cells, with an increase proportional to the course of HD [172]. In the case of EV platelets, however, mHTT was undetectable. Additionally, activated platelets no longer release EVs, indicating that mHTT-mediated procoagulant contribution is absent in HD [173]. Thus far, no study has looked at the miRNA profile carried by EVs, which could be useful in understanding and promoting HD biomarkers [174].

## 9. Multiple Sclerosis

Multiple sclerosis (MS) is a chronic autoimmune disease characterized by inflammation, demyelination, and neuro-axonal damage. It is frequently linked to progressive disability and a variety of clinical outcomes and prognoses [175]. Recently, researchers identified the miR-150-5p and let-7b-5p myeloid EVs as potential biomarkers for cognitive deficits in MS [114]. In relapsing–remitting MS (RRMS), miR-326 is upregulated in EVs derived from conventional T cells. It is associated with immunopathogenesis and may serve as a biomarker for the diagnosis and progression of MS [115]. The concentration of EVs and their miRNA load in plasma, serum, or specific cells can be used to predict response and monitor therapies [176,177,178,179].

Serum EVs miRNAs are differentially expressed in isolated syndromes and RRMS, possibly contributing as stage-specific biomarker potential [180]. During RRMS relapse, the hsa-miR-122-5p, hsa-miR-196b-5p, hsa-miR-301a-3p, and hsa-miR-532-5p in serum EVs [116], as well as hsa-miR-148-5p in erythrocyte-derived EVs [117], compared to the control group were found to be differentially expressed. Moreover, differentially expressed serum exosomal miRNAs in RRMS and progressive MS can predict disease subtypes [118].

Other EV cargo may be associated with different pathways during MS pathology, providing potential non-invasive biomarkers based on the fluids detected. Myelin proteins, synaptopodin, and synaptophysin, as well as the complement cascade and sulfatide, can be altered and found in plasma, serum, neuronal, and astrocytic-plasma EVs during the pathophysiology of MS [119,120,121]. Plasma-derived EVs released by T cells, B cells, and monocytes were elevated in untreated RRMS cases with low disability, allowing for a better understanding of immune system modifications when noticeable changes in disability or demyelinating events are not yet visible [121]. Increased levels of CNS endothelial cell-derived EVs in plasma could be a biomarker of BBB permeability and active illness in MS [123].

## 10. Epilepsy

Epilepsy, a brain disorder that involves the epileptogenic processes, is characterized by a predisposition to produce epileptic seizures [181]. EVs have the potential to be diagnostic and prognostic biomarkers for epileptogenesis and epilepsy [182]. In patients with temporal lobe epilepsy, alterations in CSF miRNAs, including exosomal miRNAs, were demonstrated in a pioneering study [183]. Another study demonstrated that the EV-associated miR3613-5p, miR-4668-5p, miR-8071, and miR-197-5p isolated from plasma may distinguish people with mesial temporal lobe epilepsy and hippocampal sclerosis from healthy controls. Furthermore, miR-8071 has been linked to seizure severity [124]. In an experimental approach, the exosomal downregulation of miR-346 and miR-331-3p was seen in the forebrain, resulting in epileptogenesis in a mouse model of chronic temporal lobe epilepsy (TLE) [184].

In TLE patients, a plasma sEVs-derived miRNA panel (hsa-miR-342-5p, hsa-miR-584-5p, hsamiR-125b-5p, hsa-miR-199a-3p, and hsa-miR-199a-5p) can predict the probability of drug-resistant children [125]. The expression of miR194-2-5p, miR15a-5p, miR-132-3p, and miR-145-5p from serum exosomes may be linked to Focal Cortical Dysplasia (FCD) and refractory epilepsy via the signaling pathways of mTOR, PI3K-Akt, p53, TGF-beta, and cell cycle control [126]. In addition, the coagulation factor IX and thrombospondin-1, which may be implicated in various epileptic physiological processes, were shown to be differentially expressed in serum-derived EVs in clinical samples and chronic epilepsy models [127].

## 11. Traumatic Brain Injury

A traumatic brain injury (TBI) is a brain injury induced by an external force that occurs suddenly [185]. The profile of proteins from EVs circulating can help to promote potential biomarkers for diagnosis, identify the temporal profile, and predict outcomes in patients with TBI [128,186]. In EVs isolated from the CSF, several proteins were associated with severe TBI. The injured human brain released more EVs, which could contain cell-to-cell communication components in both cell death signaling as well as neurodegenerative pathways [187]. Most neurofunctional proteins increased in neuron-derived EVs isolated from plasma in acute TBI, while neuropathological proteins increased in chronic and acute TBI, possibly indicating phase specificity [130]. The genes CDC2, CSNK1A1, and CTSD were upregulated in salivary EVs, which could be used as biomarkers in mild TBI (mTBI) [133].

In mTBI, the tau, A42, and IL-10 concentrations in neuronal-derived EVs isolated from plasma were higher than in controls. Post-concussive symptoms and post-traumatic stress disorder were linked to tau and IL-10 levels, respectively [131]. Sleep quality was linked to the EV levels of IL-10 and TNF in mTBI patients. mTBI patients with a significant risk of sleep difficulties have higher levels of inflammatory cytokines in their blood [132]. The EV concentrations of neurofilament light chain (NfL) from plasma are related to repetitive mTBIs, even years after the injury, with the highest levels in those with unfavorable chronic symptoms [131]. Corroborating these findings, the increased GFAP and NfL concentrations in blood-derived EVs one year after injury are higher in TBI patients than controls. They are also associated with injury severity and poor recovery outcomes [129].

In TBI, distinct miRNA profiles in EVs involve different disease-related pathways. A predictive blood test for TBI can be created using miR-203b-5p, miR-203a-3p, miR-206, and miR-185-5p EVs from plasma [134]. In the chronic mTBI cohort, 45 miRNAs in plasma EVs were substantially different from control groups, and they were mostly related to pathways involved in neuronal function, vascular remodeling, blood–brain barrier integrity, and neuroinflammation [135]. The hsa-miR-139-5p and hsa-miR-18a-5p EVs from plasma were significantly differentially expressed in the repetitive TBI and could be used to predict chronic TBI symptom persistence [188]. In TBI cases associated with disturbed consciousness, elevated GFAP and eleven miRNAs were seen in EVs from plasma. These data, taken together, could represent a valuable source of TBI biomarkers [189].

## 12. Amyotrophic Lateral Sclerosis

Amyotrophic lateral sclerosis (ALS) is a motoneuron degenerative disease, progressive and lethal [190]. Regretfully, it is still challenging to identify ALS in its early stages. EVs may be important candidates as a diagnostic tool for ALS, since it is accumulated in biological fluids. There are pathological hallmarks of ALS, including TAR DNA-binding protein 43 (TDP 43), superoxide dismutase (SOD1), fused in sarcoma (FUS), and dipeptide repeat proteins (DPRs), which are involved in the spread of the disease [191]. EVs have been shown to contain aberrant protein aggregates, altered miRNAs, and lipids involved with ALS pathology (Table 1). Studies in vitro, in vivo, and in patients with ALS have shown elevated TDP43, SOD1, FUS, and DRP levels, as well as alterations in other proteins in EVs isolated from different fluids [136,137,138,139,140,141,142,143,144,145,147,148,192]. Also, several studies described a wide range of miRNAs contained in EVs isolated from the plasma, serum, and CSF of patients with ALS [27,149,150,151,152,153]. In addition, a study showed that the total lipid content of EVs was increased in patients with ALS [154].

## 13. EVs as Treatment for CNS Diseases

After Raposo and coauthors had shown in 1996 that EVs could stimulate adaptative immune responses, several studies were undertaken to understand their role and how and if they could be used as treatment tools. At first, it was known that EVs were closely linked to the pathogenesis of some diseases, including neurological diseases such as Alzheimer’s and Parkinson’s, becoming new therapeutic targets [193,194,195]. Further, they were exploited directly as therapeutic agents, not only because they can stimulate tissue repair and act as immunotherapy agents but also because of the possibility of their use as vehicles for drug delivery [193,194,195,196]. There are some clinical trials using EVs as a treatment for CNS diseases such as epilepsy (NCT05886205), Alzheimer’s (NCT04388982), and stroke (NCT03384433).

Almost a decade after the initial understanding of the role of EVs, their ability to transfer exogenous nucleic acids and proteins between cells was brought to light [195,196]. Because they are secreted by most cells and can be acquired from the patients, they are a more biocompatible vehicle for drug delivery than other available vehicles [195]. The techniques of loading Evs with therapeutic agents are divided into two groups: exogenous loading (when the molecule of interest is incorporated into or onto the isolated EVs) and endogenous loading (when the molecules are provided so the parental cell can incorporate them into the EVs they will secrete) [194,195,196]. EVs also have the innate ability to cross the BBB, and if specific surface ligands are employed, they become a valuable treatment tool for CNS diseases [194,195,196] as shown in Figure 1. The studies’ characteristics and main findings are summarized in Table 2.

## 14. Alzheimer Disease (AD)

Known as the most prevalent kind of dementia, AD patients present neurodegeneration, and the pathogenesis of this disorder is complex and multifactorial. Fewer studies are testing EVs in AD models compared to other pathologies. Among these studies, it was shown in vitro that MSC-EVs derived from human adipose tissue contained neprilysin, an enzyme able to lyse the beta-amyloid peptide, a peptide strongly associated with the pathogenesis of AD [218,219,220].

In vivo, the administration of EVs can also be helpful in AD because of their scavenger role. They can trap the beta-amyloid peptide and deliver it to microglia, where it can be degraded, increasing its clearance [218,219]. Numerous preclinical studies also demonstrate that stem cell-derived EVs are an emerging therapeutic tool for several CNS disorders [214]. They can be loaded with many different agents and present the ability to cross the BBB, becoming an important drug vehicle [193,196,221,222]. Depending on what is loaded into the EVs, they can block gene defects, replace missing genes, modulate the downstream effects of diseases, and change cell phenotypes [222]. Thus far, studies suggest that EVs may have to be administered several times once they are biologically degraded [222]. They also demonstrated that EVs are an alternative to cell therapy because of their lesser side effects such as oncogenic transformation, undesirable cellular differentiation, and embolic risk [196]. EVs can also target both neurons and glia following intranasal administration, as demonstrated in a mouse model of AD [223] implying that EVs can potentially induce transcriptomic changes in both neurons and glia that could slow down disease pathogenesis and cognitive problems.

Despite promising prospects, there needs to be more quantitative data on the amount of EVs or drugs loaded into EVs that reach the CNS after intravenous or intranasal administration [222]. In addition, the endogenous cargo from the parental cell may cause adverse effects in some diseases or even interact with the loaded material [221]. Therefore, a better characterization of chosen EVs in preclinical models is critical for understanding their properties before employing them in treating AD [193,221].

## 15. Parkinson’s Disease (PD)

PD is also a common neurodegenerative disorder in which patients present the death of dopaminergic neurons in the substantia nigra of the brain due to elevated alpha-synuclein, brain inflammation, and oxidative stress [193,194,196,222]. One of the main goals of developing new therapeutics for PD and other neurodegenerative disorders is delivering drugs across the BBB to show their effects in the CNS [125]. Studies have already achieved the downregulation of alpha-synuclein in animal models through systemic injection of EVs from dendritic cells (DCs). These EVs displayed rabies viral glycoprotein to target brain cells and were loaded with siRNA to alpha-synuclein [194,196,198,222,224].

EVs were also used to deliver catalase, an antioxidant that is diminished in PD patients’ brains, as a means to cross the BBB. They were secreted from genetically modified macrophages that overexpressed catalase, and the results were reduced inflammation and neuroprotection in a mouse model. Studies have also proved that EVs loaded with catalase and delivered to mice intranasally showed a reduction in brain inflammation, when compared to free catalase administration [196,197,221,225,226].

The primary drug available for PD is dopamine, but its exclusive peripheric effects are well known once it cannot cross the BBB. However, when loaded in EVs, an improved functional recovery in a murine PD model without toxicity in the hippocampus, liver, spleen, or lungs was observed [196,204]. In vitro studies have showed that EVs from deciduous teeth-derived stem cells present neuroprotective potential and prevent apoptosis in 80% of dopaminergic neurons when EVs from stem cells maintained in 3D cultures were employed. However, such effects were absent with EVs from stem cells maintained in standard 2D culture conditions, revealing that even the same type of stem cell can release different EVs based on the medium and culture conditions employed [197,221,226].

## 16. Huntington’s Disease (HD)

HD is a neurodegenerative disorder caused by a specific alteration in the Huntington gene (HTT). It is dominantly inherited, and patients present with progressive cognitive impairment, involuntary movements, and psychiatric changes [194,196]. The onset is between 35 and 45 years, and death occurs approximately 15 years later, with no therapy available to treat or slow down disease progression, only to mitigate symptoms [221].

In preclinical studies, incorporating a hydrophobically modified small interfering RNA (hsiRNA) capable of enhancing stability and promoting cellular internalization into EVs targeted the wild and mutant huntingtin mRNA. It led to the dose-dependent silencing of huntingtin mRNA and protein both in vitro (primary cortical neurons culture) and in vivo (after infusion into the cerebral spinal fluid in an HD mouse model) [193,194,196,221,222]. However, the EVs were isolated from U87 glioblastoma cells, and researchers recognize that this cell line can provoke tumor formation, highlighting the importance of more preclinical studies [193,194,196].

Further investigations exploit the use of EVs loaded with microRNAs (miRNAs) in an HD mouse model: more specifically, miR-124, whose expression is downregulated in HD [193,194,196,224]. The miR-214 is highly and specifically expressed in all brain regions but the pituitary gland and has a regulatory role in CNS development and diseases, supporting adult neurogenesis [221,224]. It has been reported to slow down HD progression in transgenic mice by promoting neuronal differentiation. However, after the injection of EVs into the mouse striatum, Lee and colleagues showed modest therapeutic efficacy and no behavioral improvement, with a reduction in REST protein expression [194,196,197,221].

Additionally, Wu and coauthors tested the therapeutic potential of small interfering RNA (siRNA), molecules capable of using the RNA-induced silencing complex (RISC) to provide gene silencing. EVs expressing neuron-specific rabies viral glycoprotein (RVG) and loaded with siRNA targeting the Huntington transcript were injected intravenously to control and transgenic mice. The association with RVG and siRNA significantly reduced the transcript expression by up to 46% and 54%, depending on the transgenic mouse line (BACHD and N171-18Q, respectively) [221].

## 17. Multiple Sclerosis (MS)

MS is the most common non-traumatic cause of neurological disorders of the CNS. Patients present multifocal inflammation, demyelination, and neuronal loss. Studies have used MSC-EVs and shown improvements in motor deficits and reduced brain atrophy, following the expected reductions in plasma proinflammatory cytokines in animal models. In vivo and ex vivo, it was also proven that increased myelin levels and decreased oxidative stress could be obtained by employing EVs isolated from IFNy-stimulated DC. The administration of these IFNy-stimulated EVs to mice with experimental autoimmune encephalomyelitis (EAE) led to decreased disease evolution and spinal cord demyelination [193,194]. Similar results were seen with both intravenous and intranasal administrations [202,205].

Recent data showed positive results of using MSC-EVs in the EAE mouse model, a preclinical model of MS. Researchers injected EVs from MSCs and MSCs stimulated with IFN-ɣ and employed PBS solution and MSCs as controls. Mice were treated at Day 18, when the disease peaked, and were clinically evaluated. It was shown that the intravenous administration of EVs from MSCs stimulated by IFN-ɣ reduced the mean clinical score, demyelination, and neuroinflammation compared to the PBS control group. Also, an upregulation of CD4+ CD25+ FOXP3 + regulatory T cells within the spinal cords of mice was observed [198].

The same group showed that in vitro co-culture of EVs from MSCs stimulated by IFN-ɣ with activated peripheral blood mononuclear cells (PBMCs) reduced their proliferation and levels of proinflammatory cytokines while increasing their immunosuppressive components. Further, they sequenced RNA from IFN-ɣ EVs, which suggested they contained anti-inflammatory RNA responsible for partially inhibiting Tregs’ induction. These IFN-ɣ EVs also carry multiple anti-inflammatory and neuroprotective proteins, enlightening the possibility of a free-cell therapeutic.

Additional research groups have studied the effects of IV administration of MSCs EVs in a Theiler’s murine encephalomyelitis virus-induced demyelinating disease, and their results also showed a positive effect in improving motor deficits, reducing brain atrophy, increasing cell proliferation, and decreasing inflammatory infiltration and inflammation [204,226]. As the therapeutic preclinical properties of EVs for MS have been well explained, their preventive role is the subject of new studies. In 2021, Fleming developed in a mouse model a “vaccine-like treatment” for the antigens most related to MS. Since then, EVs have been studied as novel vehicles for vaccine delivery, and the researchers aim to facilitate antigen delivery to monocytes, DCs, macrophages, and microglia, bettering the immune tolerance [202,204].

## 18. Epilepsy

Characterized by recurrent seizures, epilepsy is a chronic disorder that may cause hippocampal injury, BBB disruption, and neurodegeneration, especially when patients evolve from a seizure episode to status epilepticus (SE). SE leads to acute neuroinflammation, which progresses into chronic neuroinflammation. EVs may modulate this pathological process, improving the cognitive function of the patients and preventing new spontaneous seizures [193,194,197,225].

Studies have used EVs from human bone marrow-derived MSCs in a mouse model of induced status epilepticus. EVs were applied intranasally (IN) twice within 24 h after SE, and the results showed a reduction in proinflammatory cytokines in the acute phase after SE and better cognitive and memory functions with neuroprotection and the maintenance of normal neurogenesis in the hippocampus [206]. However, additional studies are needed to better understand the role of EVs as a treatment against spontaneous recurrent seizures that emerge weeks or months after SE. The long-term effects of EV treatment after SE are unknown, particularly on chronic epilepsy development, or whether intermittent administrations after SE are critical to restrain chronic epilepsy development. Also, the potential of EV treatment to prevent psychiatric comorbidities is yet to be investigated. Despite these limitations, the overall results suggest that hMSC-EVs have potential as an antiepileptic therapy [225].

## 19. Traumatic Brain Injury (TBI)

Traumatic brain injury (TBI) may be caused by an ischemic insult or trauma, among other insults. Studies have shown that murine bone marrow-derived MSC-EVs injected intravenously into rats and mice with traumatic brain injury resulted in improved spatial learning, reduced neurological deficits and inflammation, and improved angiogenesis and neurogenesis [193,194,227]. EVs derived from MSCs were also applied intraperitoneally in animal models of preterm brain injury, and the results showed long-lasting cognitive improvement and the restoration of short-term myelination deficits [203].

Studies have also investigated the role of human MSC-derived EVs in a controlled cortical impact (CCI) injury model. Intravenous administration 1 h after injury suppressed proinflammatory cytokines (IL-1b) and tumor necrosis factor-alpha (TNF-a), improving spatial learning, promoting sensorimotor function recovery, and increasing vascular density after trauma. However, the cortical lesion volume was not altered. The effect was proven to be dose-dependent [197,225]. A recent study demonstrated that intranasal administration of MSC-EVs ninety minutes after a CCI is an efficient approach to target EVs into neurons and microglia into hemispheres on the ipsilateral and contralateral sides of a CCI [201]. In this study, dose-dependent effects of MSC-EVs were seen on both acute and chronic neuroinflammation and cognitive and mood impairments after TBI. An optical intranasal dose of MSC-EVs in this study restrained NLRP3 inflammasome activation in the acute phase of TBI, which was sustained in the chronic phase, leading to reduced chronic neuroinflammation. Notably, such EV treatment prevented the overactivation of p38/MAPK signaling in the chronic phase, providing a mechanistic insight into the reduced chronic neuroinflammation mediated by MSC-EVs. Furthermore, the study demonstrated better hippocampal neurogenesis on the injured cerebral hemisphere with the maintenance of standard brain-derived neurotrophic factor (BDNF)-phosphorylated cyclic AMP response element binding protein (cCREB) signaling [209].

Additional studies have shown that endothelial colony-forming cell-derived EVs can promote recovery in mice subjected to CCI. These EVs reduced brain edema and the degradation of tight-junction proteins. Such results have led to the belief that EVs have an essential role in restoring BBB integrity [197,225]. EVs derived from other stem cells, such as the adipose tissue MSC-derived EVs, were shown to have a beneficial role in CCI-induced TBI in rats through the noncoding RNA MALAT1. When analyzing the brain and spleen transcriptome, this RNA-regulated gene was found to be involved in inflammation. The study suggested that EVs first migrate to the spleen, liver, and lungs (within 1 h), and only after three days could the impact of EVs be observed on the brain. The study also implied that EVs work by inhibiting the release of peripheral macrophages and monocytes from the spleen, with fewer immune cells trespassing the damaged BBB [228].

## 20. Amyotrophic Lateral Sclerosis (ALS)

Amyotrophic lateral sclerosis (ALS) is a progressive neurodegenerative disease that affects the upper and lower motoneurons, which is accompanied by neuroinflammatory processes [190,194]; and the treatment is limited and unsatisfactory. Recent research has highlighted the potential of EVs as therapeutic agents in ALS, as they have shown promise in modulating inflammatory responses and promoting neuroprotection. ASC-exosomes have been shown to prevent oxidative damage and increase cell viability in the motoneuron-like cell line NSC-34, which has been transfected with various human superoxide dismutase (SOD1) mutations to mimic the behavior of ALS motoneurons [194]. According to Lee and colleagues, ADSCs-derived exosomes decrease elevated SOD-1 aggregation and alter cellular phenotypes in the G93A ALS in vitro model. Also, some advantageous effects of ADSC-exosomes have been demonstrated in the restoration of mitochondrial functioning [196]. Additionally, ASC-exosomes can reverse the mitochondrial dysfunction that the mutant SOD1(G93A) protein causes in NSC-34 cells [218]. In treated SOD1(G93A) mice, Bonafede and coauthors showed that the repeated injection of ASC-exosomes enhanced the motor performance; preserved lumbar motoneurons, the neuromuscular junction, and muscle; and reduced the activation of glial cells. Furthermore, exosomes can settle in areas of the animal brain with ALS lesions [219]. Human bone marrow-derived endothelial progenitor cell (hBMEPC)-derived EVs have been shown to be able to lessen the damage that ALS animal plasma causes to endothelial cells [220]. Together, these findings support the potential application of EVs in developing novel therapeutic strategies for ALS.

## 21. EVs and CRISPR-Based Gene Editing Systems

Approximately 90% of human pathogenic genetic variants are single-base mutations or insertions and deletions of fewer than a dozen base pairs [220], underscoring the relevance of genetic editing tools in addressing these genetic aberrations. By leveraging CRISPR-based approaches, researchers can generate cellular and animal models harboring disease-associated mutations, facilitating the elucidation of disease mechanisms and the screening of potential therapeutic interventions. Moreover, the advent of precise genome editing techniques, such as base editing and prime editing (reviewed in Refs. [229,230]), holds promise for correcting pathogenic mutations at the molecular level, offering potential avenues for therapeutic intervention.

In order for genome editors to fulfill their therapeutic potential within living organisms, they need to be effectively delivered to specific cells without triggering any undesirable reactions. Enhancing the delivery efficiency of CRISPR/Cas9 and gene editing is achievable through the use of viral vectors. However, the practical application of viral vectors is hindered by inherent drawbacks, including safety concerns, high expenses, complexity in handling, and limited cargo capacity [231,232,233]. Nonviral vectors, on the other hand, rely on physical methods and synthetic or natural materials, including microinjection, electroporation, liposomes, nanoparticles, and gold nanoparticles [234,235]. While these alternatives offer advantages such as simplicity, cost-effectiveness, and versatility, they are hindered by issues such as cytotoxicity, limited tissue penetration, and imprecise targeting delivery, constraining their further advancement [235,236].

Considering their capacity to carry diverse biomacromolecules, target specific cells, modulate immune responses, and undergo engineering modifications, EVs emerge as promising candidates for delivering CRISPR-based gene editors. In general, the delivery mechanism for the CRISPR/Cas9 system can be categorized into three main forms: DNA, RNA, or ribonucleoprotein complexes (RNPs).

Studies have initially demonstrated the capacity of tumor EVs to be engineered to deliver CRISPR/Cas9 plasmid DNA and RNP complexes both in vitro and in vivo [237,238,239]. Kim et al. demonstrated SKOV3 cancer cell-derived exosomes carrying a CRISPR-Cas9 plasmid and sgRNA to target the PARP1 gene in SKOV3 xenograft mice [10]. The researchers demonstrated that the treatment introduced indels in PARP1-positive SKOV3 cells, leading to apoptosis of these cells in mice with ovarian cancer. Furthermore, they illustrated that cancer cells exhibit a preference for internalizing exosomes derived from cancer cells over those produced by epithelial cells. This finding underscores how cell tropism can impact the targeting capability of EVs towards particular cell types [237].

EVs can be engineered to facilitate the encapsulation of CRISPR-Cas9 ribonucleoprotein (RNP) complexes. One approach involves employing protein tethering, wherein the Cas9 enzyme is fused with protein domains or peptides that interact with complementary domains within EVs, thus promoting encapsulation. For instance, the ARRDC1 domain, known for its involvement in plasma membrane budding and EV generation, can be utilized [240,241,242]. ARRDC1 interacts specifically with proteins containing WW domains, such as itchy E3 ubiquitin protein ligase (ITCH) [240]. To achieve encapsulation, the WW domains of ITCH are fused to the N-terminus of Cas9 [242]. Consequently, the genetically modified Cas9 is encapsulated within EVs through interactions between the WW domains and ARRDC1. This engineered Cas9 RNP, when encapsulated, demonstrated a GFP knockdown efficiency of up to 13.4% in U2OS cells [242].

The encapsulation of Cas9 ribonucleoprotein (RNP) complexes can also be facilitated through the fusion of a myristoylated peptide, such as Src kinase or Basp1 [243,244]. This peptide guides the fused protein to associate with the cytoplasmic membrane and, subsequently, the inner membrane of extracellular vesicles (EVs) [244]. For instance, a peptide (MGGKLSKKKKGYNVNDEKAKEKDKKAEGAA) was fused to the N-terminus of spCas9, resulting in increased levels of Cas9 within EVs [244]. Similarly, another peptide (GSNKSKPK), derived from the N-terminus of Src kinase, when fused to the N-terminus of Cas9, led to its myristoylation and enrichment within EVs [245]. EVs engineered with VSV-G at their membrane and encapsulating Cas9 RNP induced knockout of the eGFP reporter gene in recipient cells with an efficiency of approximately 45% in vitro [245].

Duchenne muscular dystrophy (DMD), arising from diverse mutations in the Dmd gene, stands as one of the prevalent genetic disorders, capable of diminishing the expression of the dystrophin protein [246]. Majeau et al. showed that in mdx mice with a nonsense mutation in exon 23 of the Dmd gene, the injection of serum EVs containing CRISPR/Cas9 RNPs led to the deletion of exons 23 and 24. This resulted in effective exon deletion and the restoration of dystrophin protein expression in muscle tissues compared to the control group [247]. Furthermore, in hDMD/mdx mice expressing the human Dmd gene, the Dmd genes were also susceptible to modification [248]. Another group integrated an RNA aptamer sequence (Com) into the sgRNA loop and then fused the Com-binding protein into both terminals of CD63 [249]. This Com/Com interaction facilitated the enrichment of CRISPR-Cas9 RNPs into extracellular vesicles EVs. They further showed that these EVs were capable of introducing indels in DMD/mdx mice when injected into the tibialis anterior muscle [21]. Similarly, Gee et al. introduced a novel technique named “NanoMEDIC”, where they encapsulated RNP complexes within EVs utilizing an HIV-derived Gag protein. To facilitate the recruitment of the RNP into EVs, the researchers employed a chemical-induced dimerization system known as FKBP12 and FRB. They fused the Cas9 and Gag proteins to a heterodimerizer, enabling conditional dimerization upon the addition of an inducible chemical ligand to attract the Cas9 protein into the EVs [249]. Their findings demonstrated that NanoMEDIC achieved over 90% exon-skipping efficiency in skeletal muscle cells derived from the iPS cells of DMD patients.

In summary, using EVs to deliver CRISPR-based gene editing tools offers a promising therapeutic approach for genetic disorders. EVs possess natural advantages like cell targeting and low immunogenicity, enabling the precise editing of disease-causing mutations. Preclinical studies have shown efficacy in various models, as discussed above, though challenges like scalability and optimization remain. With ongoing advancements in EV engineering and CRISPR technology, EV-mediated CRISPR gene editing holds great potential for clinical translation, offering hope for improved outcomes in genetic disorder treatment.

## 22. Conclusions and Future Perspectives

EVs found in biological fluids are attractive candidates for biomarker discovery. On the other hand, EVs naturally shed by various stem/progenitor cells are increasingly considered excellent biologics for treating neurological and neurodegenerative diseases. As discussed in this review, different cell types can release these nanosized vesicles into body fluids, providing information about parental cells’ physiological and pathological status. More studies are critical in attaining deep knowledge regarding the biological potential of EVs for diagnosing and treating neurological and neurodegenerative diseases. The points discussed in this review highlight a great potential in further extending EV analysis for future clinical applications. Particularly, brain-derived EVs in body fluids such as plasma could aid precision medicine approaches, as analysis of the composition of EVs from specific brain cell types in individual patients could help to assess the extent of neuronal dysfunction or astrocyte- and microglia-mediated neuroinflammatory signaling mechanisms in various brain disorders. Such an approach could revolutionize the diagnosis, prognosis, and treatment of various pathologies in hard-to-treat neurological and neurodegenerative diseases.

## Figures and Tables

**Figure 1 ijms-25-07371-f001:**
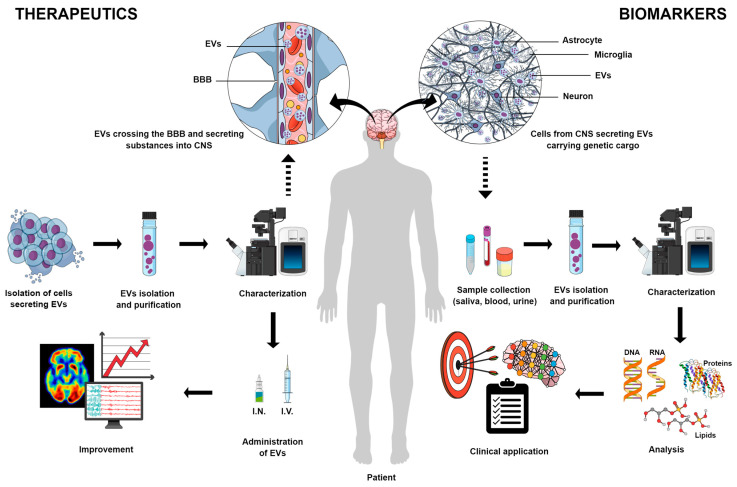
Both sides of EVs’ application: therapeutics and biomarkers. Isolated stem cells from different sources may release extracellular vesicles (EVs) with anti-inflammatory and immunomodulatory characteristics, which are significant candidates for the treatment of several neurological diseases. After EVs’ isolation, purification, and characterization by specific methods, they can be prepared for administration by intranasal (IN) or intravenous (IV) injection and the improvement of different cerebral functions can be measured. Also, EVs have the capacity to cross the blood–brain barrier (BBB), which is a fundamental characteristic for therapeutics of central nervous system (CNS) diseases. On the other side, EVs are promising biomarkers for several neurodegenerative and neurological pathologies since they are secreted by cells from CNS and can be detected in biological fluids such as saliva, blood, and urine. After EVs’ isolation, purification, and characterization and analysis of their cargo, such as DNA, RNA, proteins or/and lipids, they can be identified as biomarkers and used for different clinical applications.

**Table 1 ijms-25-07371-t001:** Main findings of extracellular vesicle biomarkers in non-invasive samples of neurological diseases. The arrows represent gene products (i.e., proteins, miRNAs, and/or both) whether they are upregulated or downregulated during disease compared to healthy controls.

Disease	Author	Year	Sample Type	EVs Type Reported	Main Findings—Biomarkers
Alzheimer’s disease	Kim, K. Y.; et al. [89]	2021	Plasma/Serum	Brain-derived exosomes	Protein	↑	Aβ-42
t-tau
p-T181-tau
p-S396-tau
Liu, W.; et al. [90]	2020	Plasma	Neuronal-derived exosomes	Protein	↓	Ng
Jia, L.; et al. [91]	2021	Plasma	Neuronal-derived exosomes	Protein	↓	Ng
GAP43
SNAP25
SYT1
Li, F.; et al. [92]	2020	Serum	Extracellular vesicles	Protein	↑	Aβ1-42
total-tau
P-T181-tau
P-S396-tau
miRNA	↑	hsa- miR-424-5p
hsa-miR-3065-5p
hsa-miR-93-5p
↓	hsa-miR-1306-5p
Durur, D. Y.; et al. [93]	2022	Plasma	Small neuronal-derived extracellular vesicles (sNDEVs)	miRNA	↑	Let-7e
Yuyama, K.; et al. [94]	2024	Plasma/CSF	Extracellular vesicles	Protein	↓	Cathepsin B
Yang, T. T.; et al. [95]	2018	Serum	Exosomes	miRNA	↑	miR-135a
miR-384
↓	miR-193b
Aharon, A.; et al. [96]	2020	Plasma (platelet-poor plasma)	Extracellular vesicles	miRNA	↓	has-let-7g-5p
has-miR126-3p
has-miR142-3p
has-miR146a-5p
has-mir26b-5p
has-mir223-3p
Cell marker	↑	CD171/L1CAM—axonal
MOG—glial
CD31+CD41− %, %CD144—endothelial
Cytokine	↑	IFN-γ
RANTES
GRO
IL-2
IL-8
AgRP
Growth Factor Content	↑	PDGF-BB
TPO
↓	UPAR
VEGF-D
Receptors VEGFR-2 and 3
FGF-4
EGF
ANG-1
Parkinson’s Disease	Shi, M.; et al. [97]	2014	Plasma	Exosomes	Protein	↑	α-syn
Stuendl, A.; et al. [98]	2021	Plasma	Extracellular vesicles	Protein	↑	α-syn
Niu, M.; et al. [99]	2020	Plasma	Neuronal-derived exosomes	Protein	↑	α-syn
Jiang, G.; et al. [100]	2020	Serum	Neuronal-derived exosomes	Protein	↑	α-syn
Chung, C-C.; et al. [101]	2021	Plasma	Extracellular vesicles	Protein	↓	α-syn
Si, X.; et al. [102]	2019	Serum	CNS-derived exosomes	Protein	↓	α-syn
Cao, Z.; et al. [103]	2019	Saliva	Exosomes	Protein	↑	α-synOlig
α-synOlig/α-syn total ratio
Ho, D. H.; et al. [104]	2014	Urine	Exosomes	Protein	↑	Protein deglycase DJ-1
Zhao, Z-H.; et al. [105]	2018	Plasma	Neural-derived exosomes	Protein	↑	Protein deglycase DJ-1
Exosomes/total protein deglycase DJ-1 ratio
α-syn
Fraser, K. B.; et al. [106]	2016	Urine	Exosomes	Protein	↑	Ser(P)-1292 LRRK2
Yao, Y-F.; et al. [107]	2018	Plasma	Exosomes	miRNA	↑	miR-331-5p
↓	miR-505
Xie, S.; et al. [108]	2022	Plasma	Extracellular vesicles	miRNA	↑	hsa-miR-30c-2-3p
↓	hsa-miR-15b-5p
hsa-miR-138-5p
hsa-miR-338-3p
hsa-miR-106b-3p
hsa-miR-431-5p
Hong, C-T.; et al. [109]	2024	Plasma	Extracellular vesicles	Proteins	↑	Synaptic proteins
Cao, X-Y.; et al. [110]	2017	Serum	Exosomes	miRNA	↑	miR-195
miR-24
↓	miR-19b
He, S.; et al. [111]	2021	Serum	Extracellular vesicles	miRNA	≠	hsa-miR-374a-5p
hsa-miR-374b-5p
hsa-miR-199a-3p
hsa-miR-28-5p
hsa-miR-22-5p
hsa-miR-151a-5p
Grossi, I.; et al. [112]	2021	Plasma	Pure small extracellular vesicles	miRNA	↑	miR-34a-5p
Chan, L.; et al. [113]	2021	Plasma	Extracellular vesicles	Cytokine	↑	pro-IL-1β (pro-interleukin)
TNF-a (tumor necrosis factor)
Multiple sclerosis	Scaroni, F.; et al. [114]	2022	Plasma	Myeloid EVs	miRNA	↑	miR-150-5p
↓	let-7b-5p
Azimi, M.; et al. [115]	2019	Whole blood	T cell-derived Exosomes	miRNA	↑	miR-326
Selmaj, I.; et al. [116]	2017	Serum	Exosomes	miRNA	↓	hsa-miR-122-5p
hsa-miR-196b-5p
hsa-miR-301a-3p
hsa-miR-532-5p
Groen, K.; et al. [117]	2020	Whole blood	Erythrocyte-derived EVs	miRNA	↑	hsa-miR-148-5p
Ebrahimkhani, S.; et al. [118]	2017	Serum	Exosomes	miRNA	↑	miR-15b-5p
miR-451a
miR-30b-5p
miR-342-3p
miR-127-3p
miR-370-3p
miR-409-3p
miR-432-5p
Moyano, A.; et al. [119]	2016	Plasma	Small Evs	Glycolipid	↑	C16:0 sulfatide
Galazka, G.; et al. [120]	2018	Serum	Exosomes	Protein	↑	MOG
Bhargava, P.; et al. [121]	2020	Plasma	Neuronal-enriched EVs	Protein	↓	Synaptopodin
Synaptophysin
Astrocytic-enriched EVs	Complement cascade component	↑	C1q
C3
C3b/iC3b
C5
C5a
Factor H
Blandford, S.; et al. [122]	2022	Plasma	Extracellular vesicles	EV concentration (cell-specific)	↑	CD3+ (T cells)
CD4+ T cells and monocytes
CD8+ T cells
CD14+ (Monocytes)
CD19+ (B cells)
Mazzucco, M.; et al. [123]	2022	Plasma	CNS endothelial-derived EVs	EV concentration	↑	CD3/CD4-pETX/MAL+ CD31+
CD3/CD4-pETX/MAL+ CD105+
CD3/CD4-pETX/MAL+ CD144+
Epilepsy	Yan, S.; et al. [124]	2017	Plasma	Exosomes	miRNA	↓	miR-4668-5p
miR-4322
miR-8071
miR-6781-5p
miR-197-5p
↑	miR3613-5p
Wang, Y.; et al. [125]	2022	Plasma	Small EVs	miRNA	↑	hsa-miR-584a-5p
hsa-miR-342a-5p
hsa-miR-150-3p
hsa-miR-125b-5p
Chen, S-D.; et al. [126]	2020	Serum	Exosomes	miRNA	↑	miR-194-2
miR-15a
miR-132
Lin, Z.; et al. [127]	2020	Serum	Exosomes	Protein	↑	F9
↓	TSP-1
Traumatic brain injury	Beard K.; et al. [128]	2021	Plasma	Extracellular vesicles	Protein	↑	GFAP
IL6
Flynn, S.; et al. [129]	2021	Serum	Extracellular vesicles	Protein	↑	GFAP
Goetzl, E. J.; et al. [130]	2019	Plasma	Neuronal-derived exosomes	Protein	↓	CD81 (Acute)
RAB10 (Acute)
↑	Annexin VII (Acute)
UCH-L1 (Acute)
All spectrin fragments (Acute)
Claudin-5 (Acute)
Occludion (Acute)
NKCC-1 (Acute)
Aquaporin 4 (Acute/Chronic)
Synaptogyrin-3 (Acute/Chronic)
Aβ-42 (Acute/Chronic)
P-T181-tau (Acute/Chronic)
P-S396 (Chronic)
total tau (Acute/Chronic)
PRPc (Acute/Chronic)
Cytokine	↑	IL-6 (Acute/Chronic)
Gill, J.; et al. [131]	2018	Plasma	Neuronal-derived exosomes	Protein	↑	tau
Aβ-42
Cytokine	↑	IL-10
Gottshall, J.; et al. [132]	2022	Plasma	Extracellular vesicles	Cytokine	↑	IL-6
Cheng, Y.; et al. [133]	2019	Saliva	Extracellular vesicles	Gene	↑	*CDC2*
*CSNK1A1*
*CTSD*
Ko, J.; et al.[134]	2020	Plasma	Brain-derived Evs	miRNA	≠	miR-203b-5p
miR-203a-3p
miR-206
miR-185-5p
Ghai, V.; et al. [135]	2020	Plasma	Extracellular vesicles	miRNA	↓	miR-139-5p

miR-143-3p
miR-146a-5p
miR-192-5p
miR-203a-3p
miR-21-5p
miR-423-5p
miR-483-5p
Amyotrophic Lateral Sclerosis	Mondola et al. [136]Gomes et al. [137] Basso et al. [138]	2003; 2007; 2013	In vitro	EVs	Protein	↑	SOD1
Iguchi et al. [139]	2016	In vitro	EVs	Protein	↑	TDP-43
Kamelgarn et al. [140]; Le Gall et al. [141]	2016; 2022	In vitro	EVs	Protein	↑	FUS
Westergard et al. [142]	2016	In vitro	EVs	Protein	↑	DPRs
Silverman et al. [143]	2019	Neural tissue	CNS-derived EVs	Protein	↑	SOD1
Ding et al. [144]	2015	CSF	SEVs	Protein	↑	TDP-43
Sproviero et al. [145]	2018	Plasma	SEVs; LEVs	Protein	↑	SOD1, TDP-43, p-TDP-43, FUS
Chen et al. [146]	2019	Plasma	LEVs	Protein	↑	IL-6
Hayashi et al. [147]	2020	CSF	SEVs	Protein	↑↓	NIR, NOC2L, PDCD6IP, VCAN, SERPINA3, PTPRZ1, C1QC, CCDC19, MYL6B, MARCO, FCGBP, FOLR1, RELN, CFB, CHMP4A
Pasetto et al. [148]	2021	Plasma	EVs	Protein	↑↓	TDP-43, HSP90, PPIA
Chen et al. [126]	2020	Plasma	SEVs	Protein	↑	TDP-43
Banack et al. [149]	2022	Plasma	EVs	miRNA	↑↓	miR-151a-5p, miR-146a-5p, miR-4454, miR-10b-5p, miR-29b-3p
Banack et al. [150]	2020	Serum	Neural-enriched EVs	miRNA	↑↓	miR-146a-5p, miR-199a-3p, miR-151a-3p, miR-151a-5p, miR-199a-5p, miR-4454, miR-10b-5p, miR-29b-3p
Saucier et al. [151]	2019	Plasma	EVs	miRNA	↑↓	miR-532-3p, miR-144-3p, miR-15a-5p, miR-363-3p, miR-183-5p miR-4454, miR-9-1-5p, miR-9-2-5p, miR-9-3-5p, miR-338-3p, miR-100-5p, miR7977, miR-1246, miR-664a-5p, miR-7641-1 miR-1290, miR-4286, miR-181a-1-5p, miR-181a-2-5p, miR-181b-1-5p, miR-181b-2-5p, miR1260b, miR-199a-1-3p, miR-199b-3p, miR-199a-2-3p, miR-127-3p, let-7c-5p
Xu et al. [152]	2018	Serum	Exosomes	miRNA	↓	miR-27a-3p
Yelick et al. [153]	2020	CSF	Exosomes	miRNA	↑	miR-124-3p
Katsu et al. [27]	2019	Plasma	Neuronal-derived EVs	miRNA	↑↓	miR-4736, miR-4700-5p, miR-1207-5p, miR-4739, miR-4505, miR-24-3p, miR-149-3p, miR-4484, miR-4688, miR-4298, miR-939-5p, miR-371a-5p, miR-3619-3p miR-1268a, miR-2861, miR-4508, miR-4507, miR-3176, miR-4745-5p, miR-3911, miR-3605-5p, miR-150-3p, miR-3940-3p, miR-4646-5p, miR-4687-5p, miR-4788,miR-4674, miR-1913, miR-634, miR-3177-3p
Morasso et al. [154]	2020	Plasma	LEVs	Lipids and metabolites	↑↓	LipidsPhenylalanine

Abbreviations: AgRP (agouti related peptide); ANG-1 (angiopoietin 1); Aβ-42 (amyloid-β42); CNS (central nervous system); CCDC19 (Coiled-coil domain-containing protein 19, mitochondrial); CFB (Complement factor B); CHMP4A (Charged multivesicular body protein 4a); C1QC (Complement C1q subcomponent subunit C); DPRs (dipeptide repeat proteins), EGF (epidermal growth factor); EVs (extracellular vesicles); FCGBP (IgG Fc-binding protein); FGF-4 (fibroblast growth factor); FOLR1 (Folate receptor alpha); FUS (fused in sarcoma), F9 (coagulation factor IX); GAP43 (growth associated protein 43); GFAP (glial fibrillary acidic protein); GRO (growth-regulated oncogene-alpha, CXCL1); IFN-γ (IFN-gamma); IL-2 (interleukin-2); IL-6 (interleukin-6); IL-8 (interleukin-8); LEVs (large extracellular vesicles); L1CAM (L1 cell adhesion molecule); MARCO (Macrophage receptor with collagenous structure); miRNA (microRNA); MOG (myelin oligodendrocyte glycoprotein); Ng (neurogranin); MYL6B (Myosin light chain 6B); NOC2L (Nucleolar complex protein 2 homolog); PDCD6IP (Programmed cell death 6-interacting protein); PDGF-BB (platelet-derived growth factor BB); PTPRZ1 (Receptor-type tyrosine-protein phosphatase zeta); p-S396-tau (tau phosphorylated at serine 396); p-T181-tau (tau phosphorylated at threonine 181); RANTES (chemokine (C-C motif) ligand 5, CCL5); RELN (Reelin); Ser(P)-1292 LRRK2 (leucine-rich repeat kinase 2); SEVs (small extracellular vesicles); SERPINA3 (Alpha-1-antichymotrypsin); SNAP25 (synaptosome associated protein 25); SOD1 (superoxide dismutase), SYT1 (synaptotagmin 1); TDP 43 (TAR DNA-binding protein 43), TPO (thrombopoietin); TSP-1 (thrombospondin-1); t-tau (total tau); UPAR (growth factor urokinase receptor); VCAN (Versican core protein); VEGF-D (vascular endothelial growth factor); α-syn (alpha-synuclein).

**Table 2 ijms-25-07371-t002:** Extracellular vesicles as therapeutic options.

Disease	Author	Year	Material	Model	Via	Main Findings
Alzheimer’s disease	Katsuda, T.; et al. [197]	2013	MSC-EVs derived from human adipose tissue containing neprilysin	In vitro	NA	Suggests the lysis of beta amyloid peptide
	Yuyama, K.; et al. [198]	2014	Neuroblastoma-derived exosomes	Heterozygotic transgenic mice that express the human APP	Stereotaxic injection	Increased clearance of beta amyloid peptite by trapping it inside the EVs and delivering it to microglica, where it can be degraded
Parkinson’s Disease	Jarmalaviciute, A.; et al. [199]	2015	EVs from stem cells from human exfoliated deciduous teeth	In vitro	NA	Neuroprotective potential and prevention of apoptosis in 80% of dopaminergic neurons when maintained in 3D culture
	Narbute, K.; et al. [200]	2019	EVs from stem cells from human exfoliated deciduous teeth	Rat model	IN	Improvement of the rat gait parameters and increased tyrosine hydroxylase expression in the substantia nigra
	Qu, M.; et al. [201]	2018	Blood EVs loaded with dopamine	Murine model	IV	Improvement of functional recovery without toxicity in hippocampus, liver, spleen, or lungs
	Cooper, J. M.; et al. [202]	2014	Exosomes expressing RVG loaded with α-Syn siRNA	Transgenic mice expressing the human phosphorylation-mimic α-Syn	IV	Downregulation of alpha synuclein
	Kojima, R.; et al. [203]	2018	HEK-293 T cells with a modified plasmid coding for catalase	Mice	IN	Reduction in brain inflammation, when compared to free catalase administration
Huntington’s Disease	Didiot, M-C.; et al. [204]	2016	Hydrophobic siRNA targeting Huntintinton mRNA in Glioblastoma EVs	Mice	Stereotactic injection in mouse striatum	Decrease in protein level in vitro and in vivo with no clinical improvement
	Liu, T.; et al. [205]	2015	EVs loaded with miR-124	R6/2 HD transgenic mice	Stereotactic injection in mouse striatum	Slowed down progression of HD, promoted neuronal differentiation and survival
	Lee, S-T.; et al. [206]	2017	miR-124 expressing HEK293 cell line	R6/2 transgenic HD mice	Stereotactic injection in mouse striatum	Reduction in REST protein expression, no behavior improvement
	Wu, T.; et al. [207]	2018	Exosomes expressing the neuron-specific RVG peptide, loaded with siRNA targeting human huntingtin exon 1	BACHD and N171-82Q transgenic mice	IV	Reduced HTT expression by up to 46% and 54%
Multiple sclerosis	Laso-Garcia, F.; et al. [208]	2018	MSC-EVs from human adipose tissue	Theiler’s murine encephalomyelitis virus-induced demyelinating disease	IV	Improvement in motor deficits, reduction in brain atrophy, increase in cell proliferation in the subventricular zone, and decrease in inflammatory infiltrates in the spinal cord
	Pusic, A. D.; et al. [209]	2013	IFNy-stimulated dendritic cells	Ex vivo (mature hippocampal slice cultures)/In vivo	NA/IN	Increase in myelin levels
	Riazifar, M.; et al. [210]	2019	Human MSCs EVs	Experimental autoimmune encephalomyelitis mouse model	IV	Stimulation by IFNγ reduced the mean clinical score of EAE mice compared to the control, reduced demyelination, decreased neuroinflammation, and upregulated the number of CD4+ CD25+ FOXP3+ regulatory T cells (Tregs) within the spinal cords of EAE mice. Co-culture of IFNγ-Exo with activated peripheral blood mononuclear cells in vitro reduced levels of proinflammatory Th1 and Th17 cytokines
Epilepsy	Long, Q.; et al. [211]	2017	MSC-EVs from human bone marrow	Mouse model of induced status epilepticus	IN	Reduction in proinflammatory cytokines, better cognitive and memory functions. Neuroprotection, reducing neural loss, preserving GABAergic intraneurons
Traumatic brain injury	Doeppner, T. R.; et al. [212]	2015	Human bone marrow-derived MSC	Mice after focal cerebral ischemia	IV	Angiogenesis, improvement of neurological impairment, long-term neuroprotection
	Zhang, Y.; et al. [213]	2015	Murine bone marrow-derived MSC-EVs	Traumatic brain injured rats and mice	IV	Improvement of spatial learning, reduction in neurological deficits, better angiogenesis and neurogenesis, reduction in inflammation
	Kim, D-K.; et al. [214]	2016	Human MSCs EVs	Traumatic brain injured mice	IV	Improvement of spatial learning and pattern separation ability, decrease in neuroinflammation
	Drommelschmidt. K.; et al. [215]	2017	Human bone marrow-derived MSC	Rodent model of inflammation-induced brain injury	IP	Improvement of long-lasting cognitive functions, amelioration of inflammation, restoration of short-term myelination deficits
	Gao, W.; et al. [216]	2018	ECFCs EVs	Traumatic brain injured mice	Stereotactic injection after trauma	Inhibition of PTEN expression and increase in AKT expression, changes accompanied by reductions in Evans blue dye extravasation, brain edema, and tight junction degradation
	Patel, N. A.; et al. [217]	2018	Adipose-derived stem cell EVs containing MALAT1	Mild controlled cortical impact-induced traumatic brain injury in rat	IV	Recovery of function on motor behavior and reduction in cortical brain injury
Amyotrophic Lateral Sclerosis	Bonafede et al. [195]	2016	Adipose-derived stem cells EVs	In vitro	NA	Decreased oxidative stress
	Lee et al. [196]	2016	Adipose-derived stem cells EVs	In vitro	NA	Recovery of mitochondrial functions
	Calabria et al. [218]	2019	Adipose-derived stem cells EVs	In vitro	NA	Recovery of mitochondrial functions
	Bonafede et al. [219]	2020	Adipose-derived stem cells EVs	SOD1 (G93A) mice	IV/IN	Enhanced motor performance, preserved lumbar motoneurons, neuromuscular junctions, and muscle, and reduced activation of glial cells
	Garbuzova-Davis and Borlongan [220]	2021	Human bone marrow-derived endothelial progenitor cell EVs	In vitro	NA	Reduced damage in endothelial cells

Abbreviations: APP (amyloid-β precursor protein); AKT (Protein kinase B); BACHD (transgenic mice expressing a neuropathogenic, full-length human mutant Huntingtin gene); EAE (experimental autoimmune encephalomyelitis); ECFCs (endothelial colony-forming cells); EVs (extracellular vesicles); HD (Huntington disease); HEK293 (immortalized human embryonic kidney cells); HTT (the gene that provides instructions for making a protein called huntingtin); IFNy (interferon gamma); IN (intranasal injection); IP (intraperitoneally injection); IV (intravenous injection); MALAT1 (metastasis-associated lung adenocarcinoma transcript 1); miR-124 (most abundant microRNA in the adult brain); mRNA (messenger RNA); MSCs (mesenchymal stem cells); NA (not applicable); N171-82Q (transgenic line expressing an N-terminally truncated human huntingtin cDNA); PTEN (protein that inhibits cell growth and promotes cellular apoptosis); REST protein (RE1-Silencing Transcription Factor); RVG (rabies viral glycoprotein); R6/2 HD (mice model expressing a portion of human Huntington’s disease gene under human gene promoter elements); siRNA (small interfering RNA); α-Syn (α-synuclein).

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
