# Peer review of "Extracellular Vesicles: The Next Generation of Biomarkers and Treatment for Central Nervous System Diseases"

_ijms, 2024, doi:10.3390/ijms25137371_

Round 1

Reviewer 1 Report

Comments and Suggestions for Authors

Extracellular Vesicles: the next generation of biomarkers and treatment for central nervous system diseases

Zanirati et. al. have explored the interesting theme of extracellular vesicles as biomarkers for nervous diseases across the diverse spectrum. This is a promising manuscript in an emerging area and is bound to draw significant interest from the research community. Here are my observations:

  1. The authors can expand on cell-cell communication and provide a little bit of extra background on this topic before going into details.
  2. The authors have drawn an excellent Figure 1.
  3. The authors can add some of the more recent findings in Table 1, which would make this table current and updated.
  4. The authors can discuss the need for better clinical methodologies to preserve the extracellular vesicles for downstream analysis.
  5. The authors need to discuss in detail the differences and errors made in the identification of vesicles (expand on all the additional validations and confirmations required for their identification/confirmation) in the shortcoming section and the critical challenges in  the practical aspects of its analysis and all the rigorous analysis.
  6. The authors can add a separate section on precision medicine implications, specifically prognostic, predictive, and diagnostic utility.
Comments on the Quality of English Language

English is mostly fine, requires minor improvements for better flow.

Author Response

Dear Reviewers,

Thank you for taking your time reviewing our study. We have revised our manuscript according to your suggestions and comments. Responses to each question or comment are inserted in the following attached letter and highlighted in the manuscript. We appreciated your review, and we followed your considerations.

Sincerely,
Gabriele Zanirati, PhD

Reviewer 1

1. The authors can expand on cell-cell communication and provide a little bit of extra background on this topic before going into details.

We added a new section entitled ‘EVs as key mediators of intercellular communication’ where we explored the role of EVs in cell-cell communication.

2. The authors have drawn an excellent Figure 1.

We sincerely appreciate the reviewer’s opinion about our figure.

3. The authors can add some of the more recent findings in Table 1, which would make this table current and updated.

More recent references were added in Table 1 and to the manuscript text.

4. The authors can discuss the need for better clinical methodologies to preserve the extracellular vesicles for downstream analysis.

We added this discussion on the ‘Conclusion and Future Perspectives’ section.

5. The authors need to discuss in detail the differences and errors made in the identification of vesicles (expand on all the additional validations and confirmations required for their identification/confirmation) in the shortcoming section and the critical challenges in the practical aspects of its analysis and all the rigorous analysis.

We discussed in the end of the ‘Background’ section the International Society for Extracellular Vesicles (ISEV) guidelines for identification of vesicles and in the ‘Conclusion and Future Perspectives’ section the critical challenges regarding analysis and reproducibility of data from EVs experiments.

6. The authors can add a separate section on precision medicine implications, specifically prognostic, predictive, and diagnostic utility.

Although the authors hold in high esteem this request from reviewer 1, we would like to emphasize that each topic for "EVs as biomarkers for CNS diseases" section contains a paragraph elucidating the prognosis and diagnosis implications for each disease included in the manuscript. However, to consistently fulfill the reviewer suggestion, we have added a paragraph to the end of section "EVs as biomarkers for CNS diseases," highlighting the points to be discussed throughout the article, providing insights into the importance of EVs as biomarkers.

Reviewer 2 Report

Comments and Suggestions for Authors

The paper reviews the role of EVs in the physiological and pathological states of the CNS and their potential use as biomarkers for diagnosis, prognosis and progression of neural diseases. It also provides information on the therapeutic use of EVs, including stem cell-derived EVs, for CNS disorders. Finally, it describes the use of EVs as vehicles for delivery of the CRISPR/Cas system for editing of mutations that cause CNS disorders.

The review is quite comprehensive, well structured, with figures and tables, which makes it easy to understand. However, I would like to suggest that the authors add some information.

1. When describing the biology of EVs, please briefly add the challenges associated with their study.

2. Although rare, amyotrophic lateral sclerosis is a real global health problem. Could you please add paragraph on the potential use of EVs in ALS treatment?   

3. When introducing EVs as therapeutic agents, please add some examples of clinical trials testing them as therapy for pathological conditions.

Author Response

Dear Reviewers,

Thank you for taking your time reviewing our study. We have revised our manuscript according to your suggestions and comments. Responses to each question or comment are inserted in the following attached letter and highlighted in the manuscript. We appreciated your review, and we followed your considerations.

Sincerely,
Gabriele Zanirati, PhD

Reviewer 2

1. When describing the biology of EVs, please briefly add the challenges associated with their study.

We added a paragraph in the “EVs as Key Mediators of Intercellular Communication” section where we discussed the main challenges associated with EVs study.

2. Although rare, amyotrophic lateral sclerosis is a real global health problem. Could you please add paragraph on the potential use of EVs in ALS treatment?

We added a paragraph on the potential use of EVs in ALS treatment, including the studies in Table 1. Furthermore, we included a paragraph related to EVs as biomarkers for ALS too.

3. When introducing EVs as therapeutic agents, please add some examples of clinical trials testing them as therapy for pathological conditions.

We added some examples of clinical trials from ClinicalTrials.gov, which use EVs as treatment and biomarkers for neurological conditions.
